# Noradrenergic deficits contribute to apathy in Parkinson's disease through the precision of expected outcomes

Frank H. Hezemans[1,2,3]*, Noham Wolpe[4,5,6], Claire O'Callaghan[6,7], Rong Ye[2], Catarina Rua[2], P. Simon Jones[2], Alexander G. Murley[2], Negin Holland[2], Ralf Regenthal[8], Kamen A. Tsvetanov[2,9], Roger A. Barker[10,11], Caroline H. Williams-Gray[10], Trevor W. Robbins[9,12], Luca Passamonti[2,13], James B. Rowe[1,2]

1 MRC Cognition and Brain Sciences Unit, University of Cambridge, Cambridge, United Kingdom, 2 Department of Clinical Neurosciences and Cambridge University Hospitals NHS Trust, University of Cambridge, Cambridge, United Kingdom, 3 Donders Institute for Brain, Cognition and Behaviour, Radboud University, Nijmegen, the Netherlands, 4 Department of Physical Therapy, The Stanley Steyer School of Health Professions, Faculty of Medicine, Tel Aviv University, Tel Aviv, Israel, 5 Sagol School of Neuroscience, Tel Aviv University, Tel Aviv, Israel, 6 Department of Psychiatry, University of Cambridge, Cambridge, United Kingdom, 7 Brain and Mind Centre and School of Medical Sciences, Faculty of Medicine and Health, University of Sydney, Sydney, Australia, 8 Division of Clinical Pharmacology, Rudolf-Boehm-Institute for Pharmacology and Toxicology, University of Leipzig, Leipzig, Germany, 9 Department of Psychology, University of Cambridge, Cambridge, United Kingdom, 10 John van Geest Centre for Brain Repair, Department of Clinical Neurosciences, University of Cambridge, Cambridge, United Kingdom, 11 Wellcome–MRC Cambridge Stem Cell Institute, University of Cambridge, Cambridge, United Kingdom, 12 Behavioural and Clinical Neuroscience Institute, University of Cambridge, Cambridge, United Kingdom, 13 Istituto di Bioimmagini e Fisiologia Molecolare, Consiglio Nazionale delle Ricerche, Milan, Italy

* frankhezemans@gmail.com

**Data Availability Statement:** Code and de-identified behavioural data and summary imaging metrics to reproduce statistical analyses, computational modelling and manuscript figures

## Abstract

Apathy is a debilitating feature of many neuropsychiatric diseases, that is typically described as a reduction of goal-directed behaviour. Despite its prevalence and prognostic importance, the mechanisms underlying apathy remain controversial. Degeneration of the locus coeruleus-noradrenaline system is known to contribute to motivational deficits, including apathy. In healthy people, noradrenaline has been implicated in signalling the uncertainty of expectations about the environment. We proposed that noradrenergic deficits contribute to apathy by modulating the relative weighting of prior beliefs about action outcomes. We tested this hypothesis in the clinical context of Parkinson's disease, given its associations with apathy and noradrenergic dysfunction. Participants with mild-to-moderate Parkinson's disease ($N$ = 17) completed a randomised double-blind, placebo-controlled, crossover study with 40 mg of the noradrenaline reuptake inhibitor atomoxetine. Prior weighting was inferred from psychophysical analysis of performance in an effort-based visuomotor task, and was confirmed as negatively correlated with apathy. Locus coeruleus integrity was assessed *in vivo* using magnetisation transfer imaging at ultra-high field 7T. The effect of atomoxetine depended on locus coeruleus integrity: participants with a more degenerate locus coeruleus showed a greater increase in prior weighting on atomoxetine *versus* placebo. The results indicate a contribution of the noradrenergic system to apathy and potential benefit from noradrenergic treatment of people with Parkinson's disease, subject to stratification according

are freely available through the Open Science Framework: https://osf.io/mry4f/.

**Funding:** The study was primarily supported by the Cambridge Centre for Parkinson-Plus to JBR, RAB and CWG (RG95450), with additional support from a Parkinson's UK research grant to JBR and CO (K-1702; https://www.parkinsons.org.uk/), a James S. McDonnell Foundation 21st Century Science Initiative Scholar Award in Understanding Human Cognition, to JBR (https://www.jsmf.org/) and an Investigator Award from the Wellcome Trust to JBR (220258; https://wellcome.org/). FHH was supported by a Cambridge Trust Vice-Chancellor's Award and Fitzwilliam College scholarship (https://www.cambridgetrust.org/). NW was supported by an Academic Clinical Fellowship (ACF-2019-14-013) from the National Institute for Health and Care Research (https://www.nihr.ac.uk/). CO was supported by a Neil Hamilton Fairley Fellowship from the Australian National Health and Medical Research Council (GNT1091310; https://www.nhmrc.gov.au/). AGM was supported by the Holt Fellowship (RG86564). NH was supported by the Association of British Neurologists, Patrick Berthoud Charitable Trust (RG99368; https://www.theabn.org/). KAT was supported by the British Academy (PF160048; https://www.thebritishacademy.ac.uk/) and the Guarantors of Brain (101149; https://guarantorsofbrain.org/). CHW-G was supported by an RCUK/UKRI Research Innovation Fellowship awarded by the UK Medical Research Council (MR/R007446/1; https://mrc.ukri.org/). JBR was supported by intramural funding (SUAG/051 G101400) and a research grant (MR/P01271X/1) from the UK Medical Research Council (https://mrc.ukri.org/). This research was supported by the NIHR Cambridge Biomedical Research Centre (BRC-1215-20014; https://cambridgebrc.nihr.ac.uk/). For the purpose of open access, the authors have applied a Creative Commons Attribution (CC BY) licence to any Author Accepted Manuscript version arising from this submission. The funders had no role in study design, data collection and analysis, decision to publish, or preparation of the manuscript.

**Competing interests:** The authors have declared that no competing interests exist.

to locus coeruleus integrity. More broadly, these results reconcile emerging predictive processing accounts of the role of noradrenaline in goal-directed behaviour with the clinical symptom of apathy and its potential pharmacological treatment.

## Author summary

Apathy is a common and harmful consequence of many neuropsychiatric diseases. Its underlying causes are not fully understood, which prevents the development of new treatments. We approach the problem in a new way, modelling human behaviour in terms of the continuously updated interaction between sensory information and brain-based predictions or 'priors' about the consequences of our actions. We have previously shown that apathy is related to a loss of precision of these 'priors'. We proposed that the precision is controlled by noradrenaline (like adrenaline, but made in the brain). We tested whether the noradrenaline-enhancing drug called atomoxetine can restore the priors' precision in apathetic people. We enrolled participants with Parkinson's disease, which is associated with both apathy and noradrenaline loss. We used ultra-high field MRI to measure individual differences in the integrity of specialist region called the locus coeruleus–the brain's source of noradrenaline. We found that the effect of treatment with atomoxetine on prior precision depended on locus coeruleus integrity: Participants with a degenerated locus coeruleus had a more positive change in prior precision. Our results highlight how individual differences in neuroanatomy can predict the potential benefit of noradrenaline treatments in people suffering from apathy.

## Introduction

Apathy is a common and debilitating feature of many neuropsychiatric diseases, including Parkinson's disease [1–3], and it occurs to varying degrees in the healthy population [4,5]. The reduction of goal-directed behaviour is typically attributed to dopamine-dependent loss of motivation [6,7], but it remains poorly understood. New treatments targeting apathy require mechanistic neurocognitive and psychopharmacological models [8].

Many psychopharmacological studies of apathy and Parkinson's disease focus on dopamine [9], relating the striatal dopamine deficit in Parkinson's disease [10] to the role of dopamine in reinforcement learning, along with value- and effort-based decision-making [11,12]. Apathetic individuals with Parkinson's disease exert less effort for a given reward, and acute withdrawal studies demonstrate a dopaminergic modulation of this effect [13–15].

However, the dopaminergic model of apathy has limitations. First, apathy is positively correlated with impulsivity, which has been attributed to hyper-dopaminergic states [5,16–19]. Second, apathy is common in Parkinson's disease patients despite dopamine replacement therapy, and may follow deep brain stimulation therapy [20]. There is no clear relationship between apathy severity and dopaminergic medication dose [1,19]. Third, evidence from animal models implicates non-dopaminergic neurotransmitter systems–in particular, noradrenaline–in motivation and effort-based decision-making [21–23].

The locus coeruleus is the principal source of noradrenaline in the brain [24,25]. In Parkinson's disease, it undergoes early and severe pathological changes [26–28], and this has been associated with certain cognitive and motivational problems that are insensitive to dopamine medication [8,29–32].

A growing body of work suggests that noradrenaline signals the uncertainty of an individual's internal model of their environment. For example, phasic bursts of noradrenaline follow salient sensory inputs and promote behavioural adaptation [33,34], while tonic release of noradrenaline correlates with higher-order contextual features such as the volatility of the environment or utility of a task, controlling the gain and selectivity of neural networks [35,36]. Evidence from pharmacological manipulations and from pupillometry data (a surrogate marker of locus coeruleus activity [37]) suggests that noradrenergic signalling indicates the extent to which sensory input is used to update existing beliefs and uncertainty of internal models [38–42]. The ability to flexibly update internal models permits adaptive engagement with the environment–supporting goal-directed, motivated behaviour. Disruption in this ability is directly relevant to apathy.

We previously proposed that apathy is a result of the dependence of motivated behaviour on the relative precision of prior beliefs about action outcomes [43]. With predictive processing, the brain optimises a probabilistic model of its environment, minimising 'surprise' or prediction error via action and perception [44,45]. The balance between active and passive (sensory) inference depends on the relative precision of prior beliefs and sensory evidence. When priors are held with high precision, they will be maintained despite conflicting (but imprecise) sensory evidence, and induce action to minimise the prediction error. That is, sensory evidence is changed by action to fulfil prior beliefs held with high precision [46–48]. If precision on the priors is not sufficiently high, passive (sensory) inference occurs by adjusting perceptual priors, and no goal-directed action is necessary.

The implication for apathy is that the loss of prior precision relative to sensory evidence would lead to a failure of action, because imprecise priors that conflict with sensory evidence are passively revised rather than actively fulfilled [47,49]. There would be an apparent 'acceptance' of the state of the world, even if discordant with goals. In support of this hypothesis, trait apathy is negatively correlated with prior precision of outcomes: more apathetic individuals have less precise prior beliefs [43]. Crucially, precision weighting has been associated with neurotransmitter systems that include noradrenaline [50–53].

This study tested the noradrenergic contributions to apathy in Parkinson's disease. We hypothesised that apathy is associated with reduced weighting of prior beliefs about action outcomes, and that this prior weighting is modulated by noradrenaline. We predicted baseline dependency of drug effects, such that the behavioural effect of a noradrenergic intervention depends on structural integrity of the locus coeruleus, in line with the inverted U-shaped dose-response curve of noradrenaline and other catecholamines [35,54–56]. We tested this hypothesis in Parkinson's disease using an effortful goal-directed visuomotor task in combination with (i) modulation of noradrenaline using atomoxetine (a noradrenergic reuptake inhibitor) and (ii) measurement of structural integrity of the locus coeruleus via ultra-high field neuromelanin-sensitive magnetic resonance imaging. In view of the potential contribution of dopamine to apathy, control analyses included the levodopa equivalent daily dose and measurement of the substantia nigra contrast as covariates.

## Results

Participants with mild-to-moderate idiopathic Parkinson's disease ($N = 17$) completed a visuomotor task that involved effortful, goal-directed behaviour (Fig 1A). Their demographic and clinical details are outlined in Table 1. They completed this task twice in a double-blind, placebo-controlled, randomised within-subjects crossover design, receiving the selective noradrenaline reuptake inhibitor atomoxetine (40 mg oral dose) or placebo. A healthy control group ($N = 20$; age-, sex- and education-matched to the Parkinson's disease group) undertook

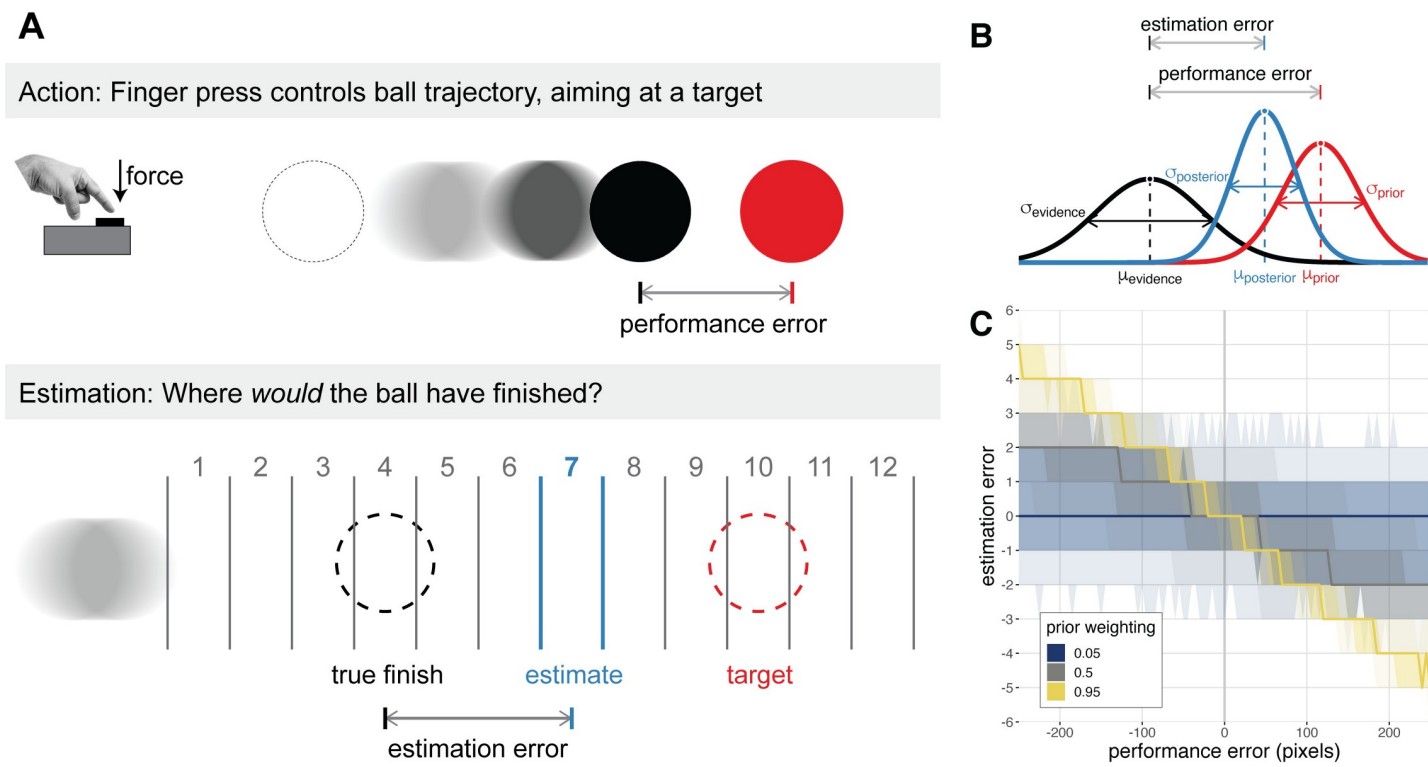

**Fig 1. Visuomotor task and Bayesian modelling approach. A)** Participants performed a sustained finger press to trigger a ballistic ball trajectory, aiming it at a target. For the majority of trials, an animation of the ball trajectory was shown, and participants sought to minimise their performance error–the distance of the final ball position from the target (top row). For the remaining trials, the ball trajectory was hidden and participants were asked to estimate where the ball would have finished. Participants were given 12 evenly spaced response options, one of which was pseudorandomly selected to be centred on the true final ball position (i.e., the veridical response option). The difference between the chosen and veridical response option constitutes the estimation error (bottom row). The pointing finger image was adapted from the Wikimedia Commons, available at https://commons.wikimedia.org/wiki/File:Index_finger_down.JPG under a CC BY-SA 3.0 license (https://creativecommons.org/licenses/by-sa/3.0/). **B)** Estimates of performance were modelled as a precision-weighted combination of a prior, centred on the target, and trial-wise sensory evidence, centred on the true final ball position. The relationship between estimation errors and performance errors is indicative of prior weighting–the precision of the prior relative to sensory evidence. **C)** Simulation demonstrating that prior weighting corresponds to the slope of the linear relationship between performance error and estimation error. For a given performance error and setting of prior weighting, the bold line represents the median predicted estimation error, and the increasingly transparent shaded areas represent the 25% and 50% quantile intervals of the predicted estimation error, respectively.

the task once, without the atomoxetine manipulation, to provide normative data. In view of baseline dependency of drug effects and the anticipated use of restorative pharmacology in patients but not healthy adults, the placebo-controlled drug intervention was restricted to those with Parkinson's disease. All participants underwent 7T MR imaging of the locus coeruleus using a magnetisation transfer weighted sequence in a baseline session, with atlas-based localisation of the locus coeruleus and contrast quantification [57]. The participants with Parkinson's disease were on their regular anti-parkinsonian medication throughout the study. Table 1 provides an overview of the participant demographics; further details about the study design and clinical characteristics of the Parkinson's disease group are provided in the Materials and Methods and S1 Text.

## Visuomotor task

The visuomotor task involved effortful, goal-directed behaviour, and was designed to estimate prior weighting, i.e. the relative precision of prior beliefs and sensory evidence in the perception of action outcomes. Using their dominant hand, participants pressed a force sensor to trigger a ballistic ball movement on the screen, aiming for the ball to stop on a target. The

**Table 1. Demographics and clinical characteristics of participants.**

| | | Parkinson's disease | Controls | BF | p |
|---|---|---|---|---|---|
| Age (years) | | 66.94 (7.29) | 65.40 (5.97) | 0.39 | .492 |
| Education (years) | | 14.24 (2.17) | 14.30 (3.29) | 0.32 | .943 |
| Male / Female | | 15 / 2 | 12 / 8 | 2.04 | .120 |
| Apathy Scale (max. 42) | | 12.68 (5.77) | 10.58 (5.09) | 0.58 | .212 |
| MMSE (max. 30) | | 29.47 (0.72) | 29.80 (0.52) | 0.87 | .127 |
| MoCA (max. 30) | | 28.00 (1.84) | 28.35 (1.50) | 0.37 | .535 |
| ACE-R | Total Score (max. 100) | 95.12 (3.72) | 97.55 (3.46) | 1.60 | .049 |
| | Attention & Orientation (max. 18) | 17.82 (0.39) | 17.95 (0.22) | 0.57 | .251 |
| | Memory (max. 26) | 23.76 (2.05) | 25.05 (1.19) | 2.68 | .031 |
| | Fluency (max. 14) | 12.18 (1.94) | 12.85 (1.76) | 0.51 | .280 |
| | Language (max. 26) | 25.82 (0.53) | 25.90 (0.45) | 0.35 | .641 |
| | Visuospatial (max. 16) | 15.65 (0.49) | 15.80 (0.70) | 0.40 | .441 |
| MDS-UPDRS | I: Nonmotor experiences (max. 52) | 8.41 (3.87) | | | |
| | II: Motor experiences (max. 52) | 12.76 (4.49) | | | |
| | III: Motor Examination (max. 132) | 27.94 (12.05) | | | |
| | IV: Motor Complications (max. 24) | 0.35 (0.79) | | | |
| | Total Score (max. 260) | 49.53 (17.55) | | | |
| Hoehn and Yahr stage (max. 5) | | 2.24 (0.44) | | | |
| Disease duration (years) | | 4.15 (1.72) | | | |
| Levodopa equivalent daily dose (mg/day) | | 659.94 (514.90) | | | |

*Note.* Data are presented as mean (SD). Group comparisons were performed with independent samples t-tests or contingency tables as appropriate. The stated *p*-values are uncorrected; none survived $p < .05$ after correction for multiple comparisons. The participants with Parkinson's disease were tested on their regular medications. Levodopa equivalent daily dose was calculated based on Tomlinson et al. [58]. Apathy Scale refers to the total score for the self-rated version of the Starkstein et al. [59] questionnaire. Abbreviations: BF, Bayes Factor for the alternative hypothesis over the null hypothesis, where $> 3$ would indicate positive evidence in favour of a group difference; MMSE, Mini Mental State Examination; MoCA, Montreal Cognitive Assessment; ACE-R, Addenbrooke's Cognitive Examination—Revised; MDS-UPDRS, Movement Disorder Society-sponsored revision of the Unified Parkinson's Disease Rating Scale.

force exerted on the sensor (relative to the participant's maximum force, measured separately) determined the initial velocity of the ball. Since the deceleration of the ball was constant, the force response (i.e., initial velocity) directly determined the ball's final position. The difference between the ball's final position and the target constituted the performance error. For 40 out of the total 120 trials, the ball's trajectory was not shown, and participants were asked to estimate where the ball would have stopped, indicating their response on a grid of 12 response options (Fig 1A; see Materials and Methods for details). The difference between the participant's selected response option and the response option that was centred on the true final ball position constituted the estimation error. The linear relationship between estimation errors and performance errors was used to infer prior weighting (Fig 1B and 1C; see Materials and Methods for details).

We first tested for differences in basic task performance between drug conditions within the Parkinson's disease group (atomoxetine vs. placebo). We also tested the effect of group (Parkinson's disease on placebo vs. controls) noting the multiple differences between these groups, including the presence of Parkinson's disease, the placebo effect, and repetition effects.

We examined to what extent the force exerted on the force sensor matched the force required to stop the ball on target (force error; see Materials and Methods for details). Among the Parkinson's disease group there was no significant effect of atomoxetine on either the median force error (Δ drug: M = 0.64%, SD = 1.84%; $t_{(16)} = 1.44$, $p = .168$; $BF = 0.60$) or the

interquartile range of force error ($\Delta$ drug: M = -0.09%, SD = 2.59%; $t_{(16)}$ = -0.14, $p$ = .892; $BF$ = 0.25). We found no significant differences between the Parkinson's disease and control groups, in terms of the median force error (controls: M = 0.69%, SD = 3.21%; PD placebo: M = -0.48%, SD = 2.78%; $t_{(34.98)}$ = 1.19, $p$ = .242; $BF$ = 0.55) and the interquartile range of force error (controls: M = 9.92%, SD = 3.73%; PD placebo: M = 9.19%, SD = 3.47%; $t_{(34.69)}$ = 0.62, $p$ = .540; $BF$ = 0.37).

## Prior weighting and apathy

We next examined participants' estimates of their own performance, to infer the relative weight afforded to prior beliefs in the perception of action outcomes. Following Bayes' rule, participants are assumed to combine their prior belief, centred on the target, with trial-wise sensory evidence, centred on the (hidden) true final ball position, to estimate the ball's final position (Fig 1B). Assuming that the prior and sensory evidence are represented as Gaussian distributions with unknown variances $\sigma^2_{\text{prior}}$ and $\sigma^2_{\text{evidence}}$, the posterior estimate for a given trial $n$ $(x^{(n)}_{\text{estimate}})$ is a precision-weighted sum of the target position $x^{(n)}_{\text{target}}$ and the true final ball position $x^{(n)}_{\text{ball}}$:

$$x^{(n)}_{\text{estimate}} = w_{\text{prior}} \cdot x^{(n)}_{\text{target}} + (1 - w_{\text{prior}}) \cdot x^{(n)}_{\text{ball}} \tag{1}$$

where the prior weighting term $w_{\text{prior}}$ is given by:

$$w_{\text{prior}} = \frac{\sigma^2_{\text{evidence}}}{\sigma^2_{\text{evidence}} + \sigma^2_{\text{prior}}} \tag{2}$$

By subtracting the true final ball position from both sides of Eq 1, we can express estimation errors as a function of performance errors:

$$\underbrace{x^{(n)}_{\text{estimate}} - x^{(n)}_{\text{ball}}}_{\text{estimation error}} = -w_{\text{prior}} \cdot \underbrace{\left(x^{(n)}_{\text{ball}} - x^{(n)}_{\text{target}}\right)}_{\text{performance error}} \tag{3}$$

Thus, the regression coefficient of estimation errors by performance errors corresponds to the negative of the prior weighting term [60]. This is illustrated in Fig 1C using simulated data: higher values of prior weighting yield estimates that are strongly drawn towards the target and away from the true final ball position. When prior weighting approaches 1, the estimation errors overwhelm the performance errors, reflecting a disregard for sensory evidence of poor performance. When prior weighting approaches 0, the estimates are in line with the true performance. We used a linear mixed effects model to estimate the relationship between estimation error and performance error, allowing the slope (i.e., negative prior weighting) to vary for each testing session for each participant. Note that this model was only fit to the subset of trials where the ball's visual trajectory was not shown (i.e., estimation trials, see Materials and Methods for details). Thus, the participants could not have used visual information to infer the ball's final position.

We confirmed that prior weighting was negatively associated with apathy, as measured by the Apathy Scale (AS)[59]. Prior weighting estimates from the Parkinson's disease group on placebo and the control group were regressed against apathy, group (Parkinson's disease vs. controls) and their interaction. To avoid confounding individual differences in prior weighting with basic task performance, we included the interquartile range of performance error as a covariate of no interest. We found a negative association between prior weighting and apathy, controlling for effects of group and performance error variability (Fig 2B; $\beta$ = -0.35, SE = 0.15,

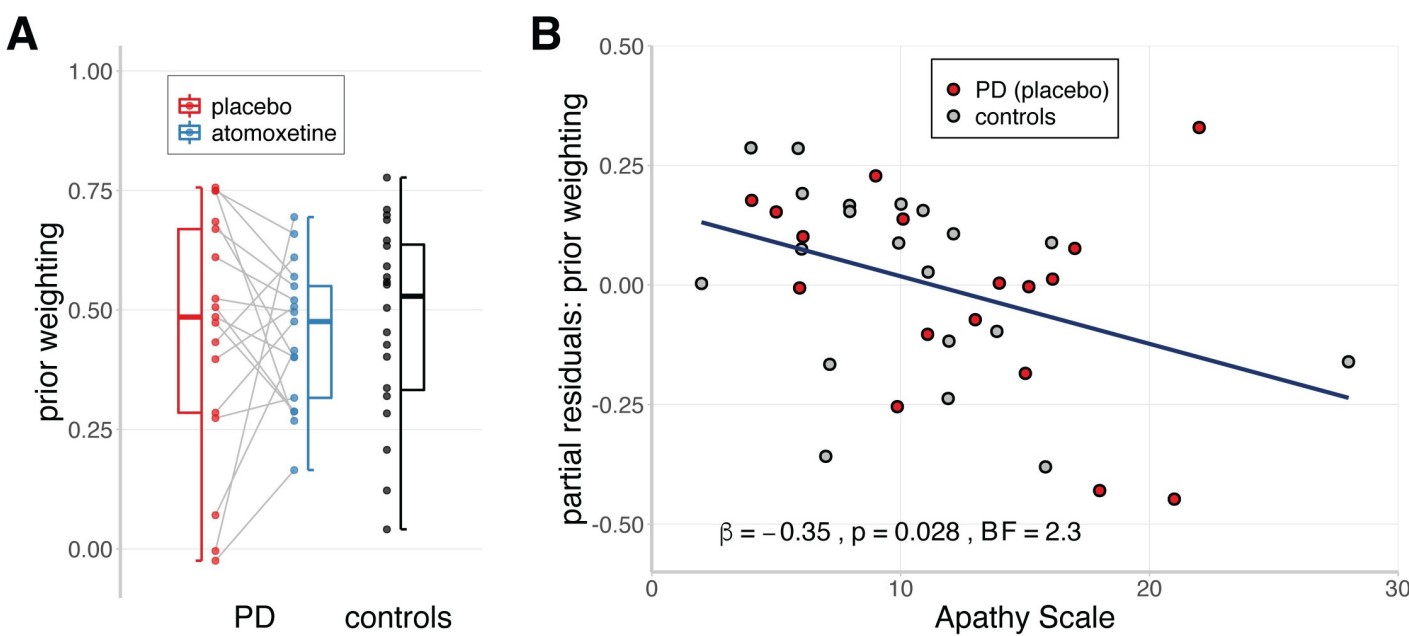

**Fig 2. Prior weighting and apathy. A)** Estimates of prior weighting, obtained through a linear mixed effects model of estimation errors against performance errors. For the Parkinson's disease group, the grey lines indicate within-subject change in prior weighting from placebo to atomoxetine. Box-plot elements: centre line, median; box limits, first and third quartiles; whiskers, most extreme observations (all within 1.5 × interquartile range from the box limits). **B)** The relationship between apathy and prior weighting, adjusted for the effects of group and task performance variability (i.e., partial residuals). Note that observations with identical questionnaire scores were horizontally jittered to avoid overlaid dots.

$t_{(32)}$ = -2.31, $p$ = .028; $BF$ = 2.30). This corroborates the negative relationship between prior weighting and trait apathy in healthy young adults [43]: more apathetic individuals show reduced prior weighting (S1 Fig). There was no group-wise difference in prior weighting between controls and patients on placebo (Fig 2A; controls: M = 0.48, SD = 0.21; PD placebo: M = 0.45, SD = 0.26; β = 0.03, SE = 0.15, $t_{(32)}$ = 0.20, $p$ = .845; $BF$ = 0.41), nor evidence for a group × apathy interaction effect (β = 0.17, SE = 0.15, $t_{(32)}$ = 1.08, $p$ = .287; $BF$ = 0.60).

## Prior weighting and noradrenaline

Having established that prior weighting was associated with apathy, we tested whether atomoxetine affected prior weighting in the Parkinson's disease group (Fig 3A). On average across the group, there was no difference in prior weighting between the atomoxetine and placebo sessions (Fig 2A; Δ drug: M = -8.50 × 10⁻⁴, SD = 0.27; $t_{(16)}$ = -0.01, $p$ = .990; $BF$ = 0.25). However, the between-subject variance in prior weighting was reduced on atomoxetine relative to placebo (Pitman's test of equality of variance for paired data: $t_{(15)}$ = -2.24, $p$ = .041; $BF$ = 2.46), which suggests a baseline-dependent drug effect [61].

We had predicted that the effect of atomoxetine depends on the structural integrity of the locus coeruleus, as indexed by its contrast to noise ratio (CNR; Fig 3B; see Materials and Methods for details). To test this, we entered prior weighting as the dependent variable in a linear mixed effects model with drug condition, locus coeruleus CNR and their interaction as fixed effects, the interquartile range of performance error as a covariate of no interest, and a random effect of participants on the intercept. We observed evidence for an interaction effect between the drug condition and locus coeruleus CNR (Fig 3C; β = -0.38, SE = 0.14, $F_{(1, 15)}$ = 7.91, $p$ = .013; $BF$ = 10.02), confirming that the effect of atomoxetine on prior weighting depended on locus coeruleus integrity. Specifically, the drug-induced change in prior weighting

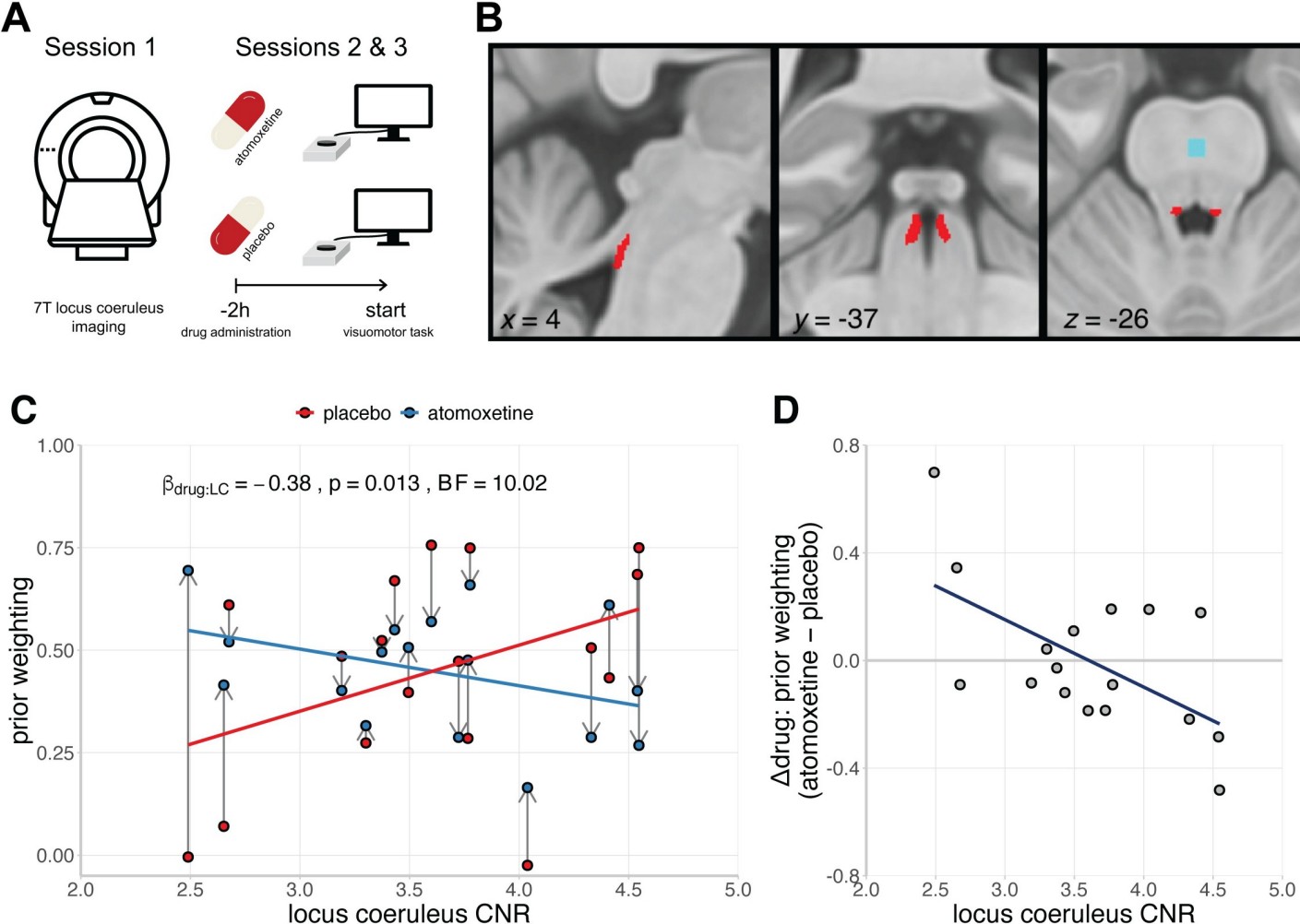

**Fig 3. Baseline-dependent effects of noradrenaline on prior weighting. A)** Schematic overview of the noradrenergic drug study. The first session involved 7T MRI of the locus coeruleus, to estimate the mean contrast-to-noise ratio (CNR). The second and third sessions formed a double-blind randomised placebo-controlled crossover study, with 40 mg of oral atomoxetine of placebo. Two hours after drug administration, participants performed the visuomotor task that was designed to estimate prior weighting. **B)** Study-specific independent locus coeruleus atlas (red) and reference region in the central pons (blue). Image reused from O'Callaghan et al. [62] under a CC-BY 4.0 license (https://creativecommons.org/licenses/by/4.0/). **C)** Estimates of prior weighting for participants with Parkinson's disease, plotted as a function of their locus coeruleus CNR and the drug condition. Within-subject change in prior weighting from placebo to atomoxetine is indicated by the grey arrows. **D)** The relationship between the drug-induced change in prior weighting (atomoxetine minus placebo) and locus coeruleus CNR.

(atomoxetine vs. placebo) was negatively associated with locus coeruleus CNR, such that participants with lower locus coeruleus CNR tended to have a more positive change in prior weighting (Fig 3D; $r_{(15)}$ = -0.59, $p$ = .013; $BF$ = 5.13). There were no main effects of the drug condition ($\beta$ = -2.06 × 10$^{-3}$, SE = 0.13, $F_{(1,\ 15)}$ = 2.36 × 10$^{-4}$, $p$ = .988; $BF$ = 0.29) or locus coeruleus CNR ($\beta$ = -0.08, SE = 0.20, $F_{(1,\ 14)}$ = 0.16, $p$ = .695; $BF$ = 0.43) on prior weighting.

Supplementary analyses confirmed the robustness and specificity of the interaction effect between the drug condition and locus coeruleus CNR. First, using both frequentist and Bayesian model selection procedures, we found that this interaction persisted with additional covariates, including age, motor severity (UPDRS part III), levodopa equivalent daily dose, atomoxetine plasma level, and total intracranial volume (S2 Text). Second, we used a linear mixed effects model with CNR extracted from the substantia nigra, as a neuromelanin-rich control region [63]. There was no interaction effect between the drug condition and substantia

nigra CNR on prior weighting ($\beta$ = 0.16, SE = 0.16, $F_{(1, 15)}$ = 0.94, $p$ = .349; $BF$ = 0.71). Third, the interaction with locus coeruleus CNR remained significant when using an alternative calculation for locus coeruleus contrast (contrast ratio to mean instead of SD; $\beta$ = -0.41, SE = 0.13, $F_{(1, 15)}$ = 9.79, $p$ = .007; $BF$ = 17.40), or when using the more conservative 25% probability mask to derive the locus coeruleus CNR ($\beta$ = -0.42, SE = 0.13, $F_{(1, 15)}$ = 10.25, $p$ = .006; $BF$ = 20.22). Fourth, re-fitting the original linear mixed effects model using a robust estimation method yielded qualitatively similar results ($\beta$ = -0.38, SE = 0.15, $t_{(15)}$ = -2.62, $p$ = .019). Taken together, these findings underwrite the robustness and specificity of the drug × locus coeruleus CNR interaction.

## Decomposition of noradrenergic effects on prior weighting

As prior weighting represents the precision of the prior relative to the sensory evidence (Eq 2), drug-induced changes in prior weighting could in principle be explained by changes in prior precision, sensory evidence precision, or both. We therefore fitted the Parkinson's disease group data with a hierarchical Bayesian model that decomposed the drug effect on prior weighting into separate drug effects on the standard deviation of the prior and sensory evidence distributions (see Materials and Methods for details). The drug effect on the standard deviation of the prior was associated with the drug effect on prior weighting (Fig 4A; $\beta$ = -0.66, SE = 0.19, $t_{(15)}$ = -3.41, $p$ = .004; $BF$ = 10.43). This relationship was negative, such that those participants who had reduced prior weighting on atomoxetine, tended to have greater standard deviation of the prior (i.e., reduced prior precision). In contrast, there was no significant relationship between the drug effect on the standard deviation of sensory evidence and the drug effect on prior weighting (Fig 4B; $\beta$ = -0.04, SE = 0.26, $t_{(15)}$ = -0.15, $p$ = .884; $BF$ = 0.42).

To directly test whether the drug effect on prior weighting was more strongly associated with the drug effect on prior precision than the drug effect on sensory evidence precision, we

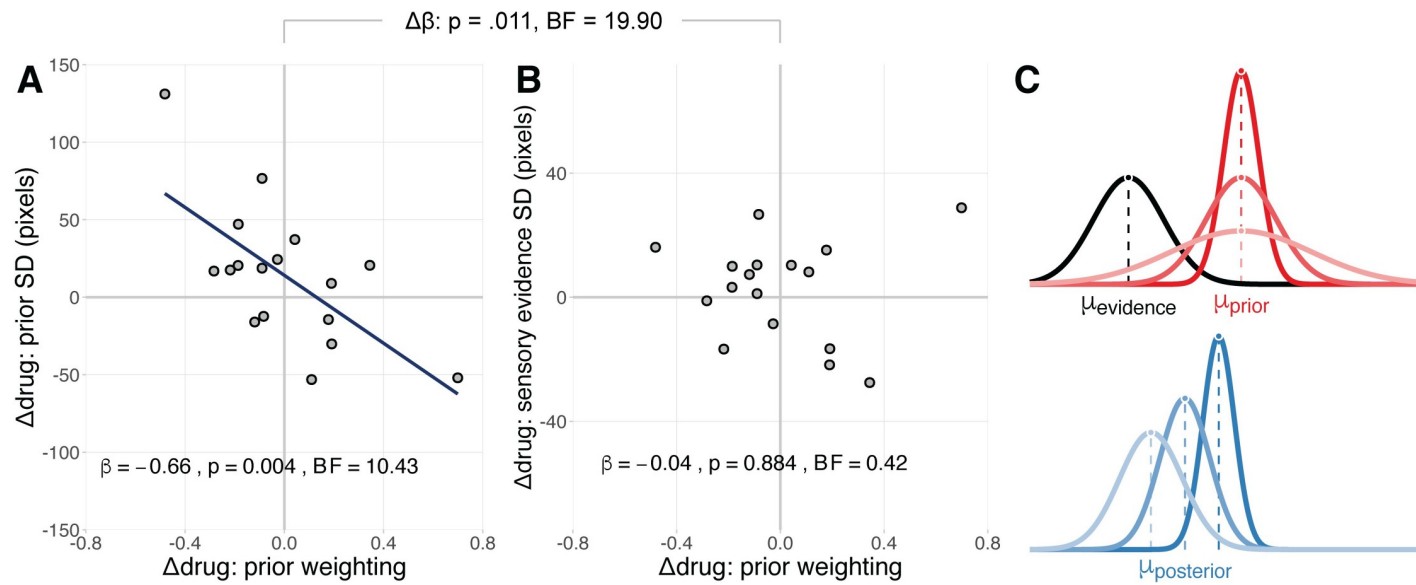

**Fig 4. Decomposition of noradrenergic effects on prior weighting. A-B)** The relationship between the drug effect on prior weighting and Bayesian model estimates of the drug effect on the standard deviation of the prior distribution (A) and the standard deviation of the sensory evidence distribution (B). **C)** Schematic illustration of how the effect of atomoxetine on prior precision (red) determines the extent to which (posterior) estimates of performance (blue) are drawn towards the prior, for a given sensory evidence (black). Lighter shades of red represent less precise priors. The effect of prior precision on the posterior is illustrated with corresponding shades of blue.

performed a repeated-measures ANCOVA with the drug effect on prior weighting as a between-subjects covariate, the precision term (prior vs. sensory evidence) as the within-subjects factor, and the estimated drug effects on prior and sensory evidence precision as the dependent variable. We found evidence for the interaction between the precision term and the drug effect on prior weighting ($F_{(1, 15)}$ = 8.34, $p$ = .011; $BF$ = 19.90), confirming that the relationships with the drug effects on prior and sensory evidence precision were significantly different from each other. We obtained similar results using a 'plausible values' analysis approach that accounts for uncertainty in the parameter estimates (S3 Text).

Overall, these results suggest that the atomoxetine-induced change in prior weighting was primarily explained by changes in prior precision, and not by changes in sensory evidence precision (Fig 4C).

## Apathy moderates noradrenergic effects on prior weighting

Lastly, given the relationship between prior weighting and apathy (Fig 2A), we tested whether the effect of locus coeruleus integrity on the atomoxetine-induced change in prior weighting varied according to the observed apathy. We regressed the drug effect on prior weighting against locus coeruleus CNR, apathy, and their interaction. We observed a significant interaction effect between locus coeruleus CNR and apathy (β = -0.49, SE = 0.21, $t_{(13)}$ = -2.34, $p$ = .036; $BF$ = 2.54), such that greater apathy was associated with a stronger (i.e. more negative) relationship between locus coeruleus CNR and atomoxetine's effect on prior weighting (Fig 5A). Specifically, the Johnson-Neyman procedure [64] indicated that the regression coefficient of locus coeruleus CNR was significant when the Apathy Scale score was above 12 (Fig 5B). This significance threshold approximates the commonly used threshold for clinically significant apathy (14 or higher)[59]. These results suggest that the baseline-dependent

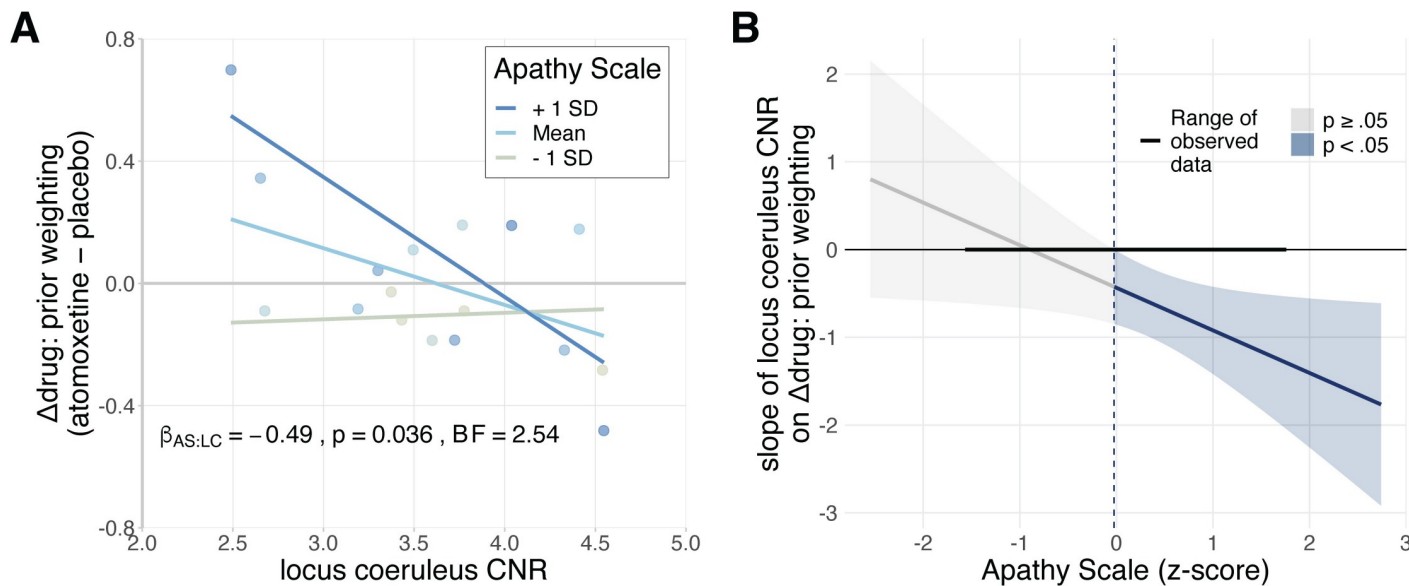

**Fig 5. Baseline-dependent noradrenergic effects on prior weighting are moderated by apathy. A)** The relationship between the drug effect on prior weighting and locus coeruleus CNR, plotted separately for above-average, average, and below-average scores on the Apathy Scale. Note that the Apathy Scale was treated as a continuous predictor in the reported statistical analyses, and is only discretised for visualisation purposes. **B)** Jonhson-Neyman interval plot, illustrating the range of scores on the Apathy Scale for which the regression coefficient of locus coeruleus CNR on the drug effect on prior weighting was statistically significant. The grey and blue shaded areas represent the 95% confidence interval for the predicted regression coefficient of locus coeruleus CNR, given the Apathy Scale score. The Apathy Scale scores (x-axis) were z-scored to facilitate interpretation of the regression coefficient (y-axis); the significance threshold of z = -0.02 (blue dashed line) corresponded to a score of 12.3.

noradrenergic effects on prior weighting is more likely to be observed among those participants who were apathetic. However, we note that the Bayes Factor of the interaction effect was "anecdotal".

## Discussion

This study provides evidence that the locus coeruleus noradrenergic system regulates the balance between prior beliefs and sensory evidence for goal-directed behaviour, and its impairment contributes to apathy. Individual differences in the relative weight afforded to prior beliefs were negatively associated with apathy, such that more apathetic individuals had reduced prior weighting. Atomoxetine modulated this prior weighting in a baseline-dependent manner in people with Parkinson's disease: people with reduced locus coeruleus integrity had a greater increase in prior weighting after atomoxetine, relative to a placebo. Using hierarchical Bayesian modelling, we demonstrated that this drug-induced change in prior weighting was primarily explained by changes in the precision of prior beliefs, and not by changes in the precision of sensory evidence. These results highlight the link between the noradrenergic locus coeruleus and the control of prior precision, during goal-directed behaviour. Given the early loss of noradrenergic cells in the locus coeruleus by Parkinson's disease, this association suggests a contributory mechanism to apathy in Parkinson's disease.

We propose that increased weighting of prior beliefs following atomoxetine may help alleviate apathy. According to Bayesian models of brain function, actions require prior beliefs about their outcomes to be held with relatively high precision [46,49]. In Parkinson's disease, the relative precision of predictive signals is compromised [65,66], which underlies the poverty of action selection and increased reliance on external cues for the initiation and maintenance of movement [67–69]. By restoring the precision of prior beliefs, atomoxetine may help attenuate disruptive sensory input [70] and minimise the unnecessary updating of prior beliefs in light of spurious prediction errors [71,72]. Thus, atomoxetine could confer a benefit to Parkinson's disease patients with reduced noradrenergic capacity, by restoring the reliance on the predictive signals that are necessary for goal-directed behaviour.

Previous studies have emphasised that the effects of catecholaminergic drugs–including atomoxetine–depend on individual differences in the baseline levels of activity of the ascending neuromodulatory systems [35,54–56]. However, without specific estimates of locus coeruleus integrity or noradrenergic capacity, heterogeneity in the response to atomoxetine is difficult to interpret [38,73]. Here, we used ultra-high field imaging to directly quantify the structural integrity of the locus coeruleus in individuals with Parkinson's disease. Our results indicate that individuals with a more severely degenerated locus coeruleus had a stronger increase in prior weighting following a single dose of atomoxetine, whereas individuals with a relatively preserved locus coeruleus had no meaningful change or even a reduction in prior weighting. These results are consistent with an inverted-U shaped curve of neurotransmitter function, whereby intermediate levels of activity are associated with optimal performance, while hypo- or hyperactive levels lead to suboptimal behaviour [35,74,75].

The relationship between locus coeruleus integrity and the drug-induced change in prior weighting varied according to apathy levels. For less apathetic individuals, there was generally no meaningful effect of atomoxetine on prior weighting, and no clear relationship between locus coeruleus integrity and the drug's effect. In contrast, among more apathetic individuals, we observed a clear baseline-dependent effect of atomoxetine on prior weighting. These results underwrite the multifactorial nature of catecholaminergic drug effects in Parkinson's disease, with complex interactions between the baseline neuromodulatory state and behavioural symptoms [54,76]. They also highlight the importance of locus coeruleus imaging for informing

noradrenergic therapy: among individuals with apathy, the cognitive effects of atomoxetine varied as a function of locus coeruleus integrity.

Our results inform theoretical models of noradrenergic function. According to the adaptive gain theory [35], dynamic shifts between predominantly phasic or tonic modes of locus coeruleus activity modulate the gain of task-relevant brain networks at different timescales, and thereby regulate the balance between task engagement and disengagement [77,78]. Phasic bursts of locus coeruleus activity, time-locked to task-relevant events, are proposed to induce an adaptive sampling bias, increasing the salience of prepotent representations while inhibiting weaker competing representations. In contrast, persistently elevated (tonic) locus coeruleus activity increases the salience of representations indiscriminately, encouraging task disengagement and exploration [35] (see also [36]). It is possible that individuals with reduced locus coeruleus integrity increase the phasic-to-tonic firing ratio following atomoxetine [79], whereas individuals with preserved locus coeruleus integrity were shifted into a predominantly tonic mode of firing. On this basis, an atomoxetine-induced increase in phasic locus coeruleus activity would cause inferences about task performance to be dominated by "optimistic" priors [60], neglecting sensory evidence for subtle deviations from this prior [80,81]. Although this interpretation of the atomoxetine-induced changes in prior weighting remains to be directly tested, it is supported by several lines of evidence. The anterior cingulate and dorsomedial frontal cortex are associated with mechanisms of belief updating [41,82–84]. Reciprocal connections between these regions and the locus coeruleus [35,85] might therefore enable noradrenergic modulation of the reliance on internal models for action [86]. In Parkinson's disease, locus coeruleus degeneration is accompanied by reduced noradrenaline levels in the forebrain [87,88]. Atomoxetine increases extracellular noradrenaline levels across the brain–including a three-fold increase in the prefrontal cortex–by inhibiting the presynaptic noradrenaline transporter [89,90]. Thus, for individuals with severe locus coeruleus degeneration, atomoxetine may help alleviate the dysfunctional modulation of prefrontal noradrenergic targets, and thereby help restore the precision of prefrontal representations of predictive signals.

This study has limitations. First, although we collected a comprehensive dataset for each individual participant, incorporating various demographic, behavioural, and neuropharmacological measures, the total number of participants is modest. Our study design was therefore not well-suited for data-driven analyses [91] or predictive modelling [92], and may have been prone to type II error for frequentist statistical inferences on small effects. We focused on the analyses of theory-driven models, with reporting of effect sizes and complementary Bayesian statistics. Second, the participants with Parkinson's disease were not severely apathetic, as the group mean score on the Apathy Scale fell below conventional cut-off scores for clinically significant apathy [1,59], and there was no statistically significant difference in apathy between the Parkinson's disease and control groups. We adopted a dimensional approach [17,93], focusing on individual differences rather than group contrasts in the mechanisms underlying apathy. Nevertheless, we recognise that it remains to be proven whether the current findings generalise to more severely apathetic cohorts. Third, we acknowledge the psychopharmacological complexity of medicated Parkinson's disease patients and atomoxetine.

Due to the limited expression of dopamine transporters in the prefrontal cortex [94], a portion of dopamine reuptake is mediated by the noradrenaline transporter [95,96]. Thus, by blocking the noradrenaline transporter, atomoxetine can increase both noradrenaline and dopamine levels in the prefrontal cortex [89,97]. It can be challenging to disentangle the noradrenergic versus dopaminergic effects of atomoxetine. Noradrenaline and dopamine are closely related neuromodulators that are synthesised from a shared metabolic pathway. They can be co-released from the same noradrenergic terminals in the prefrontal cortex [98,99]. Furthermore, locus coeruleus activity can alter midbrain dopamine cell firing [100] and

contribute to dopamine release in the hippocampus [101]. Thus, we cannot wholly exclude the possibility that atomoxetine-induced changes in prior weighting partially reflected dopaminergic effects. However, we note that the integrity of the substantia nigra yielded null results in the model, and including levodopa equivalent daily dose (LEDD) as a covariate did not change the interaction between the drug condition and locus coeruleus CNR on prior weighting. Previous studies similarly found no evidence that the effect of atomoxetine in Parkinson's disease depends on individual differences in LEDD [102]. Future studies could opt for a crossover design with both noradrenergic and dopaminergic agents, to enhance the specificity of findings.

It is possible that the changes in prior weighting are related to atomoxetine-induced changes in arousal [103,104]. However, we note that atomoxetine did not affect self-reported levels of arousal [62], nor did it affect basic task performance in the current study or in a stop-signal task, either at the group-average level or in relation to individual differences in locus coeruleus CNR [62]. This argues against a simple effect via arousal.

We used Bayesian inference as a principled framework to model the computational goal of perceptual inference in the context of effortful, goal-directed behaviour. However, we cannot comment directly on the exact algorithmic underpinnings of Bayesian perceptual inference in our experiment [105]. Following hierarchical models of predictive coding, one could decompose behaviour in our experiment into predictions at multiple levels of abstraction, from low-level proprioceptive predictions for movement to high-level multi-modal, domain-general beliefs [46,106–111]. Future work is needed to address how such a framework can accommodate the various stages of information processing that are thought to be involved in effort- and reward-based decision-making [112–114], including the weighting of action policies [13,15,53], initiating and sustaining an action [115], and evaluating and learning from the action outcomes.

We designed this study to test a mechanistic hypothesis about the noradrenergic regulation of sensorimotor integration in Parkinson's disease. It was not a clinical trial, and we did not focus on clinical outcomes. Further studies are needed to determine if atomoxetine can improve everyday functioning of people with Parkinson's disease, incorporating patient-, carer- and clinician-rated assessments as well as experimental paradigms [16,17]. Our results suggest that patients with significant apathy may benefit from noradrenergic treatment, informing future stratified clinical trials.

In conclusion, this study provides preliminary evidence for a noradrenergic role in apathy in Parkinson's disease, via the precision weighting of prior beliefs about action outcomes. We suggest that these results support a Bayesian account of apathy as a failure of active inference, resulting from impaired noradrenergic precision of priors for action. The noradrenergic modulation of prior precision may help explain dopamine-insensitive cognitive deficits in Parkinson's disease, including apathy. Locus coeruleus imaging may offer a useful marker of noradrenergic function that can inform new stratified trials of noradrenergic therapies in selected patients with neurodegenerative disease.

## Materials and methods

### Ethics statement

The study was approved by the Health Research Authority East of England–Cambridge Central Research Ethics Committee (REC 10/H0308/34), and all participants provided written informed consent in accordance with the Declaration of Helsinki.

The current study was part of a broader project on the noradrenergic mechanisms of cognitive and motivational problems in Parkinson's disease, including response inhibition deficits,

as reported in O'Callaghan et al. [62]. Therefore, the following description of the participants, study procedure, and locus coeruleus imaging overlaps with the Methods section in O'Callaghan et al. [62].

## Participants

Eighteen people with idiopathic Parkinson's disease were recruited via the University of Cambridge Parkinson's disease research clinic and through Parkinson's UK volunteer panels. All participants met the United Kingdom Parkinson's Disease Society Brain Bank criteria, were aged between 50–80 years, and had no contraindications to 7T MRI or atomoxetine. No participants had dementia, based on the Movement Disorder Society criteria for Parkinson's disease dementia [116] and the mini-mental state examination (score > 26)[117]. None had current impulse control disorders, based on clinical impression and the Questionnaire for Impulsive-Compulsive Disorders in Parkinson's Disease (QUIP-Current Short) screening tool [118]. Levodopa equivalent daily dose (LEDD) was calculated according to Tomlinson et al. [58].

Twenty-one age-, sex- and education-matched healthy controls were recruited from local volunteer panels, to provide normative data. They had no history of neurological or psychiatric disorders, and were not using psychoactive medications.

One control participant and one participant with Parkinson's disease had excessive amounts of missing visuomotor task data ($\geq$ 50% of trials) due to technical issues. After excluding these participants, the final sample consisted of 17 participants with Parkinson's disease and 20 controls. Demographic details and clinical characteristics are provided in Table 1 and S1 Text.

## Study procedure

Participants with Parkinson's disease were tested across three sessions. The first session consisted of MRI scanning and clinical assessment, including the Movement Disorder Society Unified Parkinson's Disease Rating Scale (MDS-UPDRS), mini-mental state examination (MMSE), Montreal cognitive assessment (MoCA) and the revised Addenbrooke's cognitive examination (ACE-R).

The second and third sessions formed a double-blind randomised placebo-controlled crossover study, with 40 mg of oral atomoxetine or placebo. The 40 mg atomoxetine dose is widely used as a well-tolerated 'starter dose' [119,120] that is capable of modulating behaviour and brain function in Parkinson's disease [29,102]. The drug order was pseudorandomly permuted in blocks of six successive participants, to ensure that the order was balanced across the group. This means that three participants were pseudorandomly assigned to the placebo-atomoxetine order and three to the atomoxetine-placebo order within each block of the 1st– 6th, 7th– 12th, and 13th– 18th participants. The sessions were scheduled at least 6 days apart (M = 7.29 days, SD = 1.76 days, range: 6–14 days) and at a similar time of day. For each session, blood samples were taken two hours after drug administration, to coincide with predicted peak plasma concentration of atomoxetine after a single oral dose [121]. Mean plasma concentration [122] was 264.07 ng/mL after atomoxetine (SD = 124.50 ng/mL, range: 90.92–595.11 ng/mL) and 0 ng/mL after placebo. After the blood sample, participants completed an experimental task battery that included a visuomotor task, which is the focus of the current manuscript. Participants were on their regular anti-parkinsonian medications throughout the study.

Control participants were tested in a single session that included MRI scanning and the same experimental task battery as the participants with Parkinson's disease. The control group did not undergo the drug manipulation.

All participants completed a set of questionnaires that assessed mood and various behavioural symptoms. With the exception of the Apathy Scale [59], these questionnaires were primarily collected for demographic purposes, and are described in detail in S1 Text.

## Visuomotor task

We administered a visuomotor task (adapted from Hezemans et al. [43]) that involved effortful, goal-directed behaviour, and required participants to estimate their own performance. For each trial, participants pressed on a force sensor for 3 seconds using the index finger of their dominant hand, to subsequently trigger a ballistic ball movement on the screen, aiming for the ball to stop on the target. The force response was defined as the mean force exerted from 1.5 to 2.5 seconds, divided by the participant's maximum force (estimated separately, see below). The initial velocity of the ball increased monotonically with the force response. The deceleration of the ball was kept constant. Thus, the relative force exerted on the sensor (i.e., the initial velocity) directly determined the ball's final position. As a measure of basic task performance, we calculated the force error, defined as the difference between the force response and the force required to stop the ball perfectly on target.

The task consisted of two types of trials: basic trials and estimation trials. For basic trials, participants viewed the outcome of their action–that is, the ball's full trajectory to its final position on the screen. The difference between the ball's final position and the target constituted the performance error, expressed in pixels.

**Estimating task performance.**   For estimation trials, the ball's trajectory was not shown, and participants were asked to estimate where the ball would have stopped. Participants were shown a grid of 12 evenly spaced response options (labelled with numbers 1 to 12), where one of the response options was centred on the true final ball position (i.e. the veridical response option). Participants verbally indicated their belief about the ball's final location to the experimenter as a digit ("one" to "twelve"). The difference between the selected response option and the veridical response option constituted the estimation error. Note that the target was not shown during the estimation procedure, and participants did not receive feedback regarding the true final ball position on estimation trials.

For each estimation trial, the veridical response option was selected pseudorandomly from a uniform distribution between 3 and 10. This ensured that the absolute position of the estimation grid was not indicative of the true final ball position, and that there were sufficient response options both to the left and right of the true final ball position. The width of the estimation grid was 30% of the screen width. Note that in our previous work using an unconstrained (mouse cursor) estimation procedure [43], the standard deviation of estimation error was 6.80% of the screen width.

**Task procedure.**   The task started with a practice block of 15 basic trials. In the last 5 of these practice trials, participants were asked to estimate their performance after observing the full ball trajectory, to introduce the estimation procedure. The experimenter verified that the participant understood the estimation procedure, and if necessary the practice block was repeated. The practice data was not analysed further. The subsequent test phase consisted of 4 blocks of 30 trials each. Each block consisted of 20 basic trials and 10 estimation trials. The trials within each block were pseudorandomly interleaved, with the constraints that the first 3 trials were always basic trials and that there could not be two consecutive estimation trials. In total, each testing session consisted of 120 trials, of which 40 were estimation trials. To minimise potential effects of fatigue on task performance, participants were given the opportunity to take a short break after each block.

For consistency with the original study design in Hezemans et al. [43], the task additionally featured experimental manipulations of effort and reward. Physical effort was manipulated by displaying the target either relatively close to or far from the ball's starting position, corresponding to 35% or 65% of the participant's maximum force. Reward was manipulated by either giving participants points (tallied at the top of the screen) in relation to their performance on basic trials, or not giving any points. We used a $2 \times 2$ factorial design (low effort vs. high effort; no reward vs. reward) with one block of 30 trials for each combination of effort and reward. Following the results of Hezemans et al. [43], showing minimal effects of effort and reward on prior weighting that were not associated with apathy, we report the effects of these conditions on behaviour in S4 Text.

**Maximum force calibration.** We established each participant's maximum force at the start of the task, so that the relative level of force required could be fixed across participants. Participants pressed with the maximum force they could sustain for 10 seconds. The mean force within a 5 second window with the lowest variance was taken as the response. This procedure was repeated three times, and the highest value across iterations was taken as the participant's maximum force.

## Bayesian modelling of performance estimates

The estimation trials in our visuomotor task can be considered as a Bayesian inference problem [123,124], where for each trial $n$ participants infer a hidden variable $x^{(n)}$ given noisy sensory evidence $s^{(n)}$:

$$\underbrace{p(x^{(n)}|s^{(n)})}_{\text{posterior}} \propto \underbrace{p(s^{(n)}|x^{(n)})}_{\text{likelihood}} \cdot \underbrace{p(x^{(n)})}_{\text{prior}} \qquad (4)$$

The sensory evidence $s^{(n)}$ is assumed to be the true value of the hidden variable $x^{(n)}$ corrupted by sensory noise, $s^{(n)} = x^{(n)} + \epsilon^{(n)}$, where the noise $\epsilon^{(n)}$ is sampled from a Gaussian distribution with zero mean and variance $\sigma_s^2$, that is $\epsilon^{(n)} \sim \mathcal{N}(0, \sigma_s^2)$. Thus, the likelihood is assumed to follow a Gaussian distribution with mean $x^{(n)}$ and variance $\sigma_s^2$. The prior on $x^{(n)}$ is assumed to follow a Gaussian distribution with mean $x_0^{(n)}$ and variance $\sigma_0^2$. Taken together, the posterior of $x^{(n)}$ is proportional to the product of two Gaussian distributions:

$$p(x^{(n)}|s^{(n)}) \propto \mathcal{N}(s^{(n)}; x^{(n)}, \sigma_s^2) \cdot \mathcal{N}(x^{(n)}; x_0^{(n)}, \sigma_0^2) \qquad (5)$$

The optimal estimate of the hidden variable, $\hat{x}^{(n)}$, minimises the expected loss $\mathcal{L}(\hat{x}^{(n)}, x^{(n)})$ given the sensory evidence. Since the posterior distribution in Eq 5 is Gaussian, its mean, median, and mode have the same value. We can therefore assume without loss of generality that the optimal estimate equals the posterior mean [125], which is a precision-weighted sum of the sampled sensory evidence and the prior mean:

$$\hat{x}^{(n)} = \underset{\hat{x}^{(n)}}{\operatorname{argmin}} \ \mathbb{E}[\mathcal{L}(\hat{x}^{(n)}, x^{(n)})|s^{(n)}]$$

$$= \frac{\sigma_0^2}{\sigma_0^2 + \sigma_s^2} \cdot s^{(n)} + \frac{\sigma_s^2}{\sigma_s^2 + \sigma_0^2} \cdot x_0^{(n)} \qquad (6)$$

**Estimating prior weighting.** In the current study, we assume that the sensory evidence is centred on the true final ball position $x_{\text{ball}}^{(n)}$ with variance $\sigma_{\text{evidence}}^2$, and the prior is centred on the target position $x_{\text{target}}^{(n)}$ with variance $\sigma_{\text{prior}}^2$. The observed estimate of performance $x_{\text{estimate}}^{(n)}$ can

then be modelled as the sum of $x_{\text{ball}}^{(n)}$ and $x_{\text{target}}^{(n)}$, weighted by their inverse variances:

$$x_{\text{estimate}}^{(n)} = \frac{\sigma_{\text{prior}}^2}{\sigma_{\text{prior}}^2 + \sigma_{\text{evidence}}^2} \cdot x_{\text{ball}}^{(n)} + \frac{\sigma_{\text{evidence}}^2}{\sigma_{\text{evidence}}^2 + \sigma_{\text{prior}}^2} \cdot x_{\text{target}}^{(n)} \qquad (1 \text{ restated})$$

The prior weighting term, $w_{\text{prior}} = \frac{\sigma_{\text{evidence}}^2}{\sigma_{\text{evidence}}^2 + \sigma_{\text{prior}}^2}$, can be estimated as the negative of the regression coefficient of estimation error on performance error [60]:

$$\underbrace{x_{\text{estimate}}^{(n)} - x_{\text{ball}}^{(n)}}_{\text{estimation error}} = -w_{\text{prior}} \cdot \underbrace{\left(x_{\text{ball}}^{(n)} - x_{\text{target}}^{(n)}\right)}_{\text{performance error}} \qquad (3 \text{ restated})$$

We used a linear mixed effects model to fit the linear relationship between estimation errors and performance errors, allowing the slope to vary for each of the 54 completed testing sessions (20 control participants plus 17 participants with Parkinson's disease tested twice). Prior to model fitting, the estimation errors and performance errors were z-scored for each testing session, to bring these variables onto a common scale and to ensure that the intercept was zero, as assumed by the model (Eq 3). We performed a parameter recovery analysis to ensure that this analysis procedure could reliably identify data-generating parameter values, given our experimental design and model assumptions (S5 Text).

**Hierarchical Bayesian modelling of estimation trials.**   The observed estimate of performance can alternatively be modelled probabilistically, as a sample from the full posterior distribution:

$$x_{\text{estimate}}^{(n)} \sim \mathcal{N}\left(\hat{x}^{(n)}, \sigma_{\hat{x}}^2\right) \qquad (7)$$

where the posterior variance $\sigma_{\hat{x}}^2$ is given by:

$$\sigma_{\hat{x}}^2 = \frac{\sigma_{\text{evidence}}^2 \cdot \sigma_{\text{prior}}^2}{\sigma_{\text{evidence}}^2 + \sigma_{\text{prior}}^2} \qquad (8)$$

By explicitly modelling the posterior variance, the prior and sensory evidence variances can be separately identified. For example, although an increase in prior variance or a decrease in sensory evidence variance could have the same effect on prior weighting (Eq 2), these changes would have dissociable effects on the posterior variance (Eq 8).

Potential drug effects on the prior and sensory evidence variances were modelled as atomoxetine-induced changes ($\Delta$) in these parameters, relative to the placebo session. For example, the standard deviation of the prior for participant $i$ in session $j$ was defined as follows:

$$\sigma_{\text{prior}}^{(i,j)} = \begin{cases} \sigma_{\text{prior}}^{(i)} & \text{if } j = \text{placebo} \\ \sigma_{\text{prior}}^{(i)} + \Delta_{\text{prior}}^{(i)} & \text{if } j = \text{atomoxetine} \end{cases} \qquad (9)$$

We additionally accounted for any consistent spatial shifts in the sensory evidence, $x_{\text{shift}}$, to relax the assumption that the trial-wise sensory evidence distribution was centred on the true final ball position [43,60,126]:

$$x_{\text{evidence}}^{(n)} = x_{\text{ball}}^{(n)} + x_{\text{shift}} \qquad (10)$$

The model consisted of five free parameters in total: $\sigma_{\text{prior}}$, $\sigma_{\text{evidence}}$, $\Delta_{\text{prior}}$, $\Delta_{\text{evidence}}$, and $x_{\text{shift}}$. We estimated these parameters hierarchically, such that parameters for a given

participant $i$ were sampled from corresponding group-level distributions:

$$\sigma_{\text{prior}}^{(i)} \sim \mathcal{N}_+(\mu_{\sigma_{\text{prior}}}, (\sigma_{\sigma_{\text{prior}}})^2)$$

$$\sigma_{\text{evidence}}^{(i)} \sim \mathcal{N}_+(\mu_{\sigma_{\text{evidence}}}, (\sigma_{\sigma_{\text{evidence}}})^2)$$

$$\Delta_{\text{prior}}^{(i)} \sim \mathcal{N}(\mu_{\Delta_{\text{prior}}}, (\sigma_{\Delta_{\text{prior}}})^2)$$

$$\Delta_{\text{evidence}}^{(i)} \sim \mathcal{N}(\mu_{\Delta_{\text{evidence}}}, (\sigma_{\Delta_{\text{evidence}}})^2)$$

$$x_{\text{shift}}^{(i)} \sim \mathcal{N}(\mu_{x_{\text{shift}}}, (\sigma_{x_{\text{shift}}})^2) \tag{11}$$

where $\mathcal{N}_+$ denotes a Gaussian distribution that is truncated to only allow positive values. We generally assigned relatively broad ("weakly informative") prior distributions on the group-level means and variances of the model parameters. The priors on $\mu_{\Delta_{\text{prior}}}$, $\mu_{\Delta_{\text{evidence}}}$, and $\mu_{x_{\text{shift}}}$ were all centred on zero–that is, we conservatively assumed *a priori* that on average there would be no drug effects or spatial shift. Further details about the model specification are provided in S2 Fig. Our primary interest was in the participant-level estimates of drug effects on the standard deviation of the prior and sensory evidence distributions.

We additionally fit three variants of this model, where the prior and / or sensory evidence standard deviation parameters were constrained to be fixed across the placebo and atomoxetine sessions. Specifically, one model variant only allowed for drug effects on the prior standard deviation, another model variant only allowed for drug effects on sensory evidence standard deviation, and a final model variant did not include any drug-induced change parameters. However, a comparison of each model's estimated pointwise predictive accuracy [127] favoured the inclusion of drug effects on both the prior and sensory evidence standard deviations (S3 Fig), and we therefore focused on the parameter estimates from the 'full' model.

We used Markov Chain Monte Carlo (MCMC) sampling to estimate the posterior distributions of the model parameters. We used 8 chains with 4000 samples each, and discarded the first 2000 samples of each chain as the warm-up. Model convergence was confirmed by the potential scale reduction statistic $\hat{R}$ ($< 1.004$ for all parameters), and by visual inspection of the time-series plots of the MCMC samples. The model's goodness of fit was assessed by visually comparing the observed data to simulated data generated from the model's posterior predictive distribution (S4 and S5 Figs). For a given parameter of interest, we took the median of its posterior distribution as the optimal estimate.

## MRI acquisition

The MR images were acquired with a 7T Magnetom Terra (Siemens, Erlangen, Germany), using a 32-channel head coil (Nova Medical, Wilmington, USA). The locus coeruleus was imaged using a 3D magnetisation transfer (MT) weighted sequence at high resolution [57,128]. The sequence included 112 oblique, axial slices oriented perpendicular to the long axis of the brainstem, to cover both the midbrain and the pontine regions. A train of 20 Gaussian-shaped RF pulses was applied at 6.72 ppm off resonance, 420° flip angle, followed by a turbo-flash readout (TE = 4.08 ms, TR = 1251 ms, flip-angle = 8°, voxel size = 0.4 x 0.4 x 0.5 $mm^3$, 6/8 phase and slice partial Fourier, bandwidth = 140 Hz/px, no acceleration, 14.3%-oversampling, TA ~ 7 min). The transmit voltage was adjusted for each participant based on the average flip angle in the central area of the pons, which was obtained from a B1 pre-

calibration scan. The MT scan was repeated twice and averaged offline to enhance the signal-to-noise ratio. An additional scan was acquired with the same parameters as above but without the off-resonance pulses. For anatomical coregistration, a high resolution T1-weighted structural image (0.7 mm isotropic) was acquired using the MP2RAGE sequence with the UK7T Network harmonised protocol: TE = 2.58 ms, TR = 3500 ms, BW = 300 Hz/px, voxel size = 0.7 x 0.7 x 0.7 mm$^3$, FoV = 224 x 224 x 157 mm$^3$, acceleration factor (A>>P) = 3, flip angles = 5/2˚ and inversion times (TI) = 725/2150 ms for the first/second images.

## Image processing and locus coeruleus integrity

The image processing pipeline is described in detail in O'Callaghan et al. [62], which used an identical pipeline as the current study (based on Ye et al. [57]). The MT images were bias field corrected, and then entered into a T1-driven coregistration pipeline to warp the images to the isotropic 0.5 mm ICBM152 (International Consortium for Brain Mapping) T1-weighted asymmetric template [129].

The co-registered MT images were used to quantify the contrast with respect to a reference region, the central pontine tegmentum, generating contrast-to-noise ratio (CNR) maps. For each participant $i$ and voxel $j$, the signal $x^{(i,j)}$ was contrasted with the mean reference signal $\mu_{\text{ref}}^{(i)}$, and then divided by the standard deviation of the reference signal $\sigma_{\text{ref}}^{(i)}$:

$$\text{CNR}^{(i,j)} = \frac{x^{(i,j)} - \mu_{\text{ref}}^{(i)}}{\sigma_{\text{ref}}^{(i)}} \tag{12}$$

To ensure that the CNR values were localised to the locus coeruleus, we used an independent locus coeruleus atlas based on a separate sample of 29 age- and education-matched healthy controls [62]. For each axial slice on the rostrocaudal extent, the locations of the left and right locus coeruleus were determined using a semi-automated segmentation method [57]. The locus coeruleus voxels were segmented into binary images, and then averaged and thresholded at 5% to obtain a template (705 voxels, 88.125 mm$^3$; see Supplementary Materials in O'Callaghan et al. [62]).

To estimate the locus coeruleus integrity, we applied the independent locus coeruleus atlas to each participant's CNR map, and then calculated the mean CNR value across the whole structure. We also performed this calculation after applying a more stringent locus coeruleus atlas that was thresholded at 25% (274 voxels, 34.25 mm$^3$). In addition, we performed an alternative calculation of locus coeruleus contrast that replaced the denominator in Eq 12 with the mean reference signal: $\text{CR}^{(i,j)} = (x^{(i,j)} - \mu_{\text{ref}}^{(i)})/\mu_{\text{ref}}^{(i)}$, yielding a contrast ratio rather than contrast-to-noise ratio. However, as described in the Results section, these alternative contrast methods did not meaningfully change the relationship between locus coeruleus integrity and the atomoxetine-induced change in prior weighting. Comparisons of locus coeruleus CNR between the Parkinson's disease and control groups are reported in detail in O'Callaghan et al. [62].

We used an analogous approach to obtain CNR from the substantia nigra, as a neuromelanin-rich control region. This analysis pipeline is described in detail in the Supplementary Materials in O'Callaghan et al. [62] and in Rua et al. [63]. In brief, the MT images from the separate sample of 29 age- and education-matched healthy controls were used to create an independent probabilistic atlas of the substantia nigra, thresholded at 5%. This atlas was then applied to the current study sample to compute a CNR map (Eq 12), where the reference region was defined as the midbrain background (*crus cerebri*). The mean CNR value for each participant served as the estimate of substantia nigra CNR.

## Statistical inference

We report both frequentist and Bayes factor (BF) analyses for hypothesis testing, with a significance threshold of $p$ = .05 (two-sided) for frequentist analyses. We present the BF for the alternative hypothesis over the null hypothesis (i.e., $BF_{10}$), such that $BF > 3$ indicates "positive evidence" for the alternative hypothesis. All BF analyses used the default 'JZS' prior on the effect size under the alternative hypothesis. To obtain BFs for specific effects in ANOVAs and linear (mixed) models, we used Bayesian model averaging to estimate the change from prior to posterior inclusion odds ("inclusion BF"). This BF indicates how much more likely the data are when a given effect is included in the model, compared to when the effect is excluded [130]. The $p$-values for fixed effects in linear mixed models were obtained using the Kenward-Roger approximation. Prior to analyses, we z-scored all continuous variables, and assigned sum-to-zero contrasts to categorical variables.

## Software and equipment

The visuomotor task was implemented in MATLAB R2018b using the Psychophysics Toolbox extensions version 3 [131], and was displayed on a 12.5-inch laptop screen ($1920 \times 1080$ pixels). The force sensor had a sampling rate of 60 Hz and a measurement accuracy of ±9.8 mN. The magnetisation transfer images were processed using the Advanced Normalization Tools version 2.2.0 [132] and in-house MATLAB scripts [57,62]. All statistical analyses were implemented in R version 3.6.1 [133]; detailed information about the specific R packages used for the analyses is provided in S6 Text. The hierarchical Bayesian modelling was implemented in Stan [134].

## Supporting information

**S1 Fig. Relationship between prior weighting and apathy in young adults.** Data from Hezemans et al. [43], demonstrating the relationship between trait apathy (measured using the Apathy Motivation Index) and prior weighting, adjusted for task performance variability (i.e., partial residuals). Note that observations with identical questionnaire scores were horizontally jittered to avoid overlaid dots. Full statistics for the regression coefficient of apathy: β = -0.42, SE = 0.14, $t(44)$ = -3.02, $p$ = .004; BF = 11.63.
(PDF)

**S2 Fig. Plate notation and sampling statements for the hierarchical Bayesian model.** Participant-level parameters were sampled from latent group-level distributions. We allowed for atomoxetine-induced changes in the standard deviations of the prior and sensory evidence distributions. Session-level parameters were then used to obtain trial-level posterior distributions of the final ball position. The observed estimates of performance were modelled as samples from the trial-level posterior distributions. The model is represented in plate notation: shaded nodes represent observed data whereas white nodes represent latent variables; rectangular nodes represent discrete or fixed variables whereas circular nodes represent continuous variables; and double-bordered white nodes represent deterministic variables whereas single-bordered white nodes represent stochastic variables.
(PDF)

**S3 Fig. Information criteria for variants of the hierarchical Bayesian model.** In addition to the model presented in the manuscript, we fit three variants of the model with restrictions on the number of drug-induced change parameters. For each model variant, we computed the leave-one-out information criterion (LOOIC) and the widely applicable information criterion (WAIC) as estimates of the model's expected predictive accuracy. For both measures, the 'full'

model which included drug-induced change parameters for both the prior and sensory evidence standard deviations was strongly preferred over the three more restricted model variants.
(PDF)

**S4 Fig. Posterior predictive check: response distributions.** Each panel compares the observed responses (light-coloured density plot and histogram) to distributions of simulated responses drawn from the model's posterior predictive distribution (dark-coloured density traces).
(PDF)

**S5 Fig. Posterior predictive check: predictive error.** Each panel illustrates the distribution of the mean predictive error of the model–that is, the observed responses minus simulated responses drawn from the model's posterior predictive distribution, averaged across Markov Chain Monte Carlo samples. These histograms can therefore be interpreted as the distributions of residuals.
(PDF)

**S1 Text. Mood and behaviour questionnaires. Table A. Descriptive statistics and group comparisons of questionnaires.** Data are presented as mean (SD). Group comparisons were performed with independent samples t-tests. The stated p-values are uncorrected; none survived p < .05 after correction for multiple comparisons. Abbreviations: BF, Bayes Factor for the alternative hypothesis over the null hypothesis, where > 3 would indicate positive evidence in favour of a group difference; BIS, Barratt Impulsiveness Scale; HADS, Hospital Anxiety and Depression Scale; MEI, Motivation and Energy Inventory; CAARS, Conners' Adult ADHD Rating Scale; RBDSQ, REM sleep Behaviour Disorder Screening Questionnaire; CBI-R, Cambridge Behavioural Inventory–Revised. **Fig A. Density plots of questionnaire scores for participants with Parkinson's disease (blue) and controls (orange).** The questionnaire scores were z-scored to bring the different questionnaires onto a common scale (note that this transformation does not affect group comparisons for a given questionnaire outcome). Tick marks reflect individual data points. Abbreviations: BIS, Barratt Impulsiveness Scale; MEI, Motivation and Energy Inventory; HADS, Hospital Anxiety and Depression Scale; CAARS, Conners' Adult ADHD Rating Scale.
(DOCX)

**S2 Text. Role of covariates in the drug × LC CNR interaction on prior weighting. Table A. Backward elimination of fixed effects in the linear mixed effects model predicting prior weighting.** Values for predictors are standardised regression coefficients (ß). $^*p < .05$. Drug, atomoxetine vs. placebo condition; LC CNR, Locus Coeruleus Contrast to Noise Ratio; ICV, total intracranial volume; Ato plasma, atomoxetine plasma concentration; LEDD, Levodopa Equivalent Daily Dose; Session, first vs. second session; UPDRS III, Unified Parkinson's Disease Rating Scale, motor examination; AIC, Akaike Information Criterion; BIC, Bayesian Information Criterion; Δ AIC / BIC, difference in AIC / BIC with respect to the lowest AIC / BIC value. All models included a fixed effect of the interquartile range of performance error as a covariate of no interest, and a random effect of participants on the intercept. Total intracranial volume was estimated from the T1-weighted MP2RAGE images using the mri_segstats–etiv-only procedure in FreeSurfer v6.0.0. **Table B. Bayes Factors for the inclusion of fixed effects in the linear mixed effects model predicting prior weighting.** P(incl): prior inclusion probability, i.e. the summed prior probability of models that include the predictor. A priori, all possible restrictions of the full model were deemed to be equally likely (i.e., a uniform prior was assigned to the model space). Thus, P(incl) reflects the proportion of alternative models

that included the predictor. P(incl|data): posterior inclusion probability, i.e. the summed posterior probability of models that include the predictor. P(excl|data): posterior exclusion probability, i.e. the summed posterior probability of models that exclude the predictor. $BF_{inclusion}$: Inclusion Bayes Factor, i.e. the change from prior to posterior inclusion odds. This indicates how much more likely the data are under models that include the predictor, compared to models that exclude the predictor. This analysis was performed using "matched" models, which means that (i) models were not permitted to include an interaction effect without its constituent main effects, and (ii) inclusion probabilities for an interaction effect were based only on the subset of models that contained (at least) the constituent main effects of the interaction. All models included a fixed effect of the interquartile range of performance error as a covariate of no interest, and a random effect of participants on the intercept.
(DOCX)

**S3 Text. Decomposing drug effects on prior weighting: Plausible values analysis. Fig A. Decomposing the noradrenergic effects on prior weighting using a plausible values analysis approach.** (A-B) Distributions of plausible correlations between the drug effect on prior weighting and the estimated drug effect on the standard deviation of the prior (A) or sensory evidence (B). (C) Distribution of the difference between the plausible correlations.
(DOCX)

**S4 Text. Effects of effort and reward on task performance and prior weighting. Fig A. Task accuracy (median force error) plotted as a function of effort, reward, group, and drug.** Dots represent individual participants, and boxplots represent the marginal distribution for a given condition. For the Parkinson's disease group, the grey lines indicate within-subject change in median force error from placebo to atomoxetine. Box-plot elements: centre line, median; box limits, first and third quartiles; whiskers, most extreme observations within $1.5 \times$ interquartile range from the box limits. **Fig B. Task variability (interquartile range of force error) plotted as a function of effort, reward, group, and drug.** Dots represent individual participants, and boxplots represent the marginal distribution for a given condition. For the Parkinson's disease group, the grey lines indicate within-subject change in the interquartile range of force error from placebo to atomoxetine. Box-plot elements: centre line, median; box limits, first and third quartiles; whiskers, most extreme observations within $1.5 \times$ interquartile range from the box limits. **Fig C. Prior weighting plotted as a function of effort, reward, group, and drug.** Dots represent individual participants, and boxplots represent the marginal distribution for a given condition. For the Parkinson's disease group, the grey lines indicate within-subject change in prior weighting from placebo to atomoxetine. Box-plot elements: centre line, median; box limits, first and third quartiles; whiskers, most extreme observations within $1.5 \times$ interquartile range from the box limits. **Table A. Estimated marginal means of median force error by group, effort, and reward. Table B. Estimated marginal means of median force error by drug, effort, and reward. Table C. Estimated marginal means of interquartile range of force error by group, effort, and reward. Table D. Estimated marginal means of interquartile range of force error by drug, effort, and reward. Table E. Estimated marginal means of prior weighting by group, effort, and reward. Table F. Estimated marginal means of prior weighting by drug, effort, and reward.**
(DOCX)

**S5 Text. Parameter recovery of prior weighting. Fig A. Parameter recovery of prior weighting.** (A) Data-generating prior weighting plotted against the estimated (i.e., recovered) prior weighting. The diagonal (identity) line represents perfect parameter recovery. For a given data-generating prior weighting value, the dot represents the median of the prior weighting

estimates across 2000 simulations of estimation errors; the vertical error bars represent the 95% quantile intervals of the prior weighting estimates. The inset histogram illustrates the distribution of the difference between the median estimated prior weighting and data-generating prior weighting. (B, C) Examples of data simulations for a relatively low data-generating value of prior weighting (B; orange dot and error bar in A) and a relatively high data-generating value of prior weighting (C; green dot in A).

(DOCX)

**S6 Text. Details of statistical software.**
(DOCX)

## Acknowledgments

We thank the volunteers for their participation, staff at the Wolfson Brain Imaging Centre and National Institute for Health and Care Research (NIHR) Cambridge Clinical Research Facility for their help with data collection, and members of the Cambridge Centre for Frontotemporal Dementia and Related Disorders for valuable suggestions and discussions.

The views expressed are those of the authors and not necessarily those of the National Institute for Health and Care Research or the Department of Health and Social Care.

## Author Contributions

**Conceptualization:** Frank H. Hezemans, Noham Wolpe, Claire O'Callaghan, Trevor W. Robbins, James B. Rowe.

**Data curation:** Frank H. Hezemans, Claire O'Callaghan, Rong Ye, Catarina Rua.

**Formal analysis:** Frank H. Hezemans, Noham Wolpe, Claire O'Callaghan, Rong Ye, Catarina Rua, P. Simon Jones, Ralf Regenthal.

**Funding acquisition:** Claire O'Callaghan, Roger A. Barker, Caroline H. Williams-Gray, James B. Rowe.

**Investigation:** Frank H. Hezemans, Claire O'Callaghan, Rong Ye, Catarina Rua, Alexander G. Murley, Negin Holland.

**Methodology:** Frank H. Hezemans, Noham Wolpe, Claire O'Callaghan, Rong Ye, Catarina Rua, P. Simon Jones, James B. Rowe.

**Project administration:** Frank H. Hezemans, Claire O'Callaghan, Rong Ye, James B. Rowe.

**Resources:** Ralf Regenthal, Roger A. Barker, Caroline H. Williams-Gray, James B. Rowe.

**Software:** Frank H. Hezemans, Noham Wolpe, Claire O'Callaghan, Rong Ye, Catarina Rua, P. Simon Jones, Kamen A. Tsvetanov.

**Supervision:** Noham Wolpe, Claire O'Callaghan, Luca Passamonti, James B. Rowe.

**Visualization:** Frank H. Hezemans, Claire O'Callaghan, Rong Ye.

**Writing – original draft:** Frank H. Hezemans.

**Writing – review & editing:** Frank H. Hezemans, Noham Wolpe, Claire O'Callaghan, Rong Ye, Catarina Rua, P. Simon Jones, Alexander G. Murley, Negin Holland, Kamen A. Tsvetanov, Roger A. Barker, Caroline H. Williams-Gray, Trevor W. Robbins, Luca Passamonti, James B. Rowe.

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
