## [Decision Letter · Decision Letter 0]

9 Dec 2021

Dear Dr. Hezemans,

Thank you very much for submitting your manuscript "Noradrenergic deficits contribute to apathy in Parkinson's disease through the precision of expected outcomes" for consideration at PLOS Computational Biology. As with all papers reviewed by the journal, your manuscript was reviewed by members of the editorial board and by three independent reviewers. The reviewers appreciated the attention to an important topic. Based on the reviews, we are likely to accept this manuscript for publication, providing that you modify the manuscript according to the review recommendations.

Sincerely,

Woo-Young Ahn

Associate Editor

PLOS Computational Biology

Samuel Gershman

Deputy Editor

PLOS Computational Biology

[LINK]

Reviewer's Responses to Questions

**Comments to the Authors:**

Reviewer #1: This is an interesting study in which the authors use a clever task that allows them to distinguish subjects' precision of their prior expectations (about a target's movement) from the precision of the sensory likelihood (from the applied force to the target). They collect data from n=17 Parkinson's patients and controls; the former are also given Atomoxetine and placebo in a within-subject manipulation, and 7T imaging of their Locus Coereleus. The authors found a negative correlation between prior/likelihood weighting and apathy, and an interaction such that subjects with lower LC integrity showed a positive effect of ATO on this weighting (higher LC integrity resulted in a negative effect). They are careful to control for other factors, which do not affect the results.

Overall the topic is interesting and clinically important, and the methods are rigorous. There are some weaknesses: the main ones being that the sample size is fairly small (n=17), controls did not receive ATO or MRI scanning, Atomoxetine also has effects on DA as well as NA, and the PD patients were not any more apathetic than controls. This makes it difficult to draw firm conclusions about mechanisms of apathy in PD specifically. But the relationship of ATO effects on prior precision to LC integrity is certainly of interest. More detailed comments below:

p15 - The authors are careful to say that these findings "suggest" a contributory mechanism to apathy in PD, but I think they should also explicitly stress that these PD patients were not apathetic - their apathy scores were no different to controls'. So this makes it difficult to extrapolate the findings to pathological levels of apathy. I would therefore rephrase the subsequent conclusion "this study provides evidence for a noradrenergic role in apathy in Parkinson’s disease".

p3/16 - There seems to be a tension between some of the evidence about noradrenaline (NA). At the bottom of p3, lots of evidence is cited that NA relates to (phasic) unexpected uncertainty or (tonic) volatility in the environment, i.e. evidence that the model must be updated - hence priors should be weakened, in both cases. But how does this square with the finding that ATO increases the precision of priors, in those with a less intact LC? There is some rather intricate speculation in the Discussion about phasic/tonic LC firing ratios. But to me this indicates that the subjects with *intact* LC show the most intuitive result: that ATO *reduces* prior precision (as NA ought to if it encodes unexpected uncertainty). This makes me wonder whether the opposite ATO effects in those with less intact LC are in fact dopaminergic effects, unmasked by the impaired LC system? This makes more sense to me than positing an inverted-U or altered phasic/tonic relationships. Impossible to tell from this work of course, but I think this possibility should be mentioned a bit more clearly in line 430 or earlier.

p22 - Can the authors explain a bit more why the 2x2 effort/reward manipulation was included if it was not being used? They just state "we did not analyse these factors further,

following Hezemans et al. [43]" but this is a bit thin. E.g. why not check for an effect of Atomoxetine on these factors? Is there insufficient power with n=17 subjects?

Small points

p4 - I would frame the opposite of 'active' inference as perceptual inference, not 'passive' inference.

Reviewer #2: I am grateful for the opportunity to review this excellent paper. I very much enjoyed reading it, and think this is a great example of computational neurology. It is difficult to find much to improve on, but there are a few points that would benefit from clarification:

1. I thought the method for estimating the prior weighting based upon the slope between the two forms of error was very convincing in its simplicity. However, it would be useful to clarify whether this slope was calculated only for those trials in which the trajectory was not shown. I will assume this is the case in what follows. The reason this is important is that the sensory evidence is different between the conditions where the trajectory is and is not shown, and the performance error from those when it is cannot really be inserted into Equation 3, which computes the estimation error on the assumption of no visual trajectory data (the estimation will clear be different when visual data are available).

2. Following from the above, could the author's comment on the sensory modalities assumed to contribute to the evidence? If not visual, I presume this will largely be tactile and proprioceptive data associated with the finger press. If so, this might be something the authors could make more of. As their focus is on a neurological condition often characterised as a movement disorder, it seems a very good experimental design that focuses on sensory modalities that result from self-generated movements.

3. One difficulty is that the premise of this study is based upon apathy in Parkinson's disease, but it seems that the Parkinson's patients were no more apathetic than the controls. However, the authors highlight the association between locus coeruleus degeneration and apathy, and the interaction between prior weighting and the effect of pharmacological manipulations. Do the authors feel that this experiment could be practically adapted as a clinical test to identify potential responders to atomoxetine from a clinically heterogenous group?

4. In Equation 5 - presumably the authors mean N(x^(n),sigma_s^2), not N(s^(n),sigma_s^2) for the likelihood? The likelihood should be a function of the latent parameters (i.e., the parameters described by the prior and posterior).

Reviewer #3: Hezemans et al. present a study of 17 PD and 20 controls. The task was as per a previous study. Participants exerted a force to move a circle by a given distance on the screen, sometimes without feedback. When feedback is absent, they estimate how far they think they pushed the ball.

The study shows that motivated people were biased to perceive their exerted force as being closer to the target.

PD patients were given atomoxetine, which didn't change performance overall. However patients with MRI evidence of locus coeruleus degeneration showed stronger effects of drug on increasing the bias.

The result is timely and novel. Noradrenergic degeneration in PD is of great clinical and scientific interest. Although the sample size is not huge, it appears to be a very clean sample of PD patients, and the analyses are rigorous, Bayesian, and well controlled. An impressive range of control analyses are presented.

I had no major criticisms but I have a some comments on the interpretation.

1) Comparison with other models of motivation.

The framing of apathy in terms of active inference seems to make qualitatively different predictions to standard econometric theories. I think these should be highlighted, as many readers may not make the connections. In particular, the authors talk about "effort", but in their theory, there seems to be no asymmetry between exerting too much or too little force. Further, it seems a conceptual leap that a sensory prior that produces action, should translate to a prior over *perception* of action. Can the authors clarify why that should be? In general, I would appreciate more of an interface with existing theories of effort and apathy. Also, are the authors talking about physical or cognitive effort here?

Accordingly an important negative seems to be that PD patients did not exert less force than healthy people, and that noradrenaline did not increase force (and perhaps apathy did not correlate with force either?). Does this inform theories of apathy?

Further, the experimental design (effort x reward) was not analysed as such. Although this aligns with the previous paper, I believe that the reader would be genuinely interested in the effects of those manipulations, especially if they are null results. If the authors do not wish to report these, then are the data at least available?

2) Intro - NA is presented as definitively signalling uncertainty. While there are reasonable arguments for this, I think some caution is wise in some of the stronger statements eg L88.

3) I think "optimal" is used in an unusual way (page 8) - the authors write about the optimal integration, but this seems odd since the optimal prior actually has zero-weight (no precision). This is because the sensory input is not biased away from the target (I assume?) -- so it should always be "better" to use raw sensory information rather than integrate it with the target. People of course do mix the prior with evidence, but it is confusing to call this optimal.

4) I would prefer if some of the task details in Materials & Methods appeared up in the main text e.g. how the force determined the distance moved, the proportion of trials

L225: a healthy

**Have the authors made all data and (if applicable) computational code underlying the findings in their manuscript fully available?**

Reviewer #1: Yes

Reviewer #2: Yes

Reviewer #3: **No: **Did not see it in text - maybe I missed it.

PLOS authors have the option to publish the peer review history of their article (what does this mean?). If published, this will include your full peer review and any attached files.

Reviewer #1: No

Reviewer #2: No

Reviewer #3: **Yes: **Sanjay Manohar

Figure Files:

Data Requirements:

Reproducibility:

References:

---

## [Decision Letter · Decision Letter 1]

5 Apr 2022

Dear Dr. Hezemans,

We are pleased to inform you that your manuscript 'Noradrenergic deficits contribute to apathy in Parkinson's disease through the precision of expected outcomes' has been provisionally accepted for publication in PLOS Computational Biology.

Best regards,

Woo-Young Ahn

Associate Editor

PLOS Computational Biology

Samuel Gershman

Deputy Editor

PLOS Computational Biology

Reviewer's Responses to Questions

**Comments to the Authors:**

Reviewer #1: The authors have addressed all my concerns and I have no further comments. Compliments to them!

Reviewer #2: I am grateful to the authors for their responses, and have no further edits to suggest. I am happy to endorse publication at this stage.

Reviewer #3: The authors have responded to ask my comments most thoroughly. They should be congratulated on an excellent paper.

**Have the authors made all data and (if applicable) computational code underlying the findings in their manuscript fully available?**

Reviewer #1: Yes

Reviewer #2: Yes

Reviewer #3: Yes

PLOS authors have the option to publish the peer review history of their article (what does this mean?). If published, this will include your full peer review and any attached files.

Reviewer #1: No

Reviewer #2: No

Reviewer #3: **Yes: **Sanjay Manohar

---

## [Editor Report · Acceptance letter]

4 May 2022

PCOMPBIOL-D-21-01786R1 

Noradrenergic deficits contribute to apathy in Parkinson's disease through the precision of expected outcomes

Dear Dr Hezemans,

I am pleased to inform you that your manuscript has been formally accepted for publication in PLOS Computational Biology. Your manuscript is now with our production department and you will be notified of the publication date in due course.

With kind regards,

Olena Szabo
